# NRF2 and Mitochondrial Function in Cancer and Cancer Stem Cells

**DOI:** 10.3390/cells11152401

**Published:** 2022-08-04

**Authors:** Emiliano Panieri, Sónia A. Pinho, Gonçalo J. M. Afonso, Paulo J. Oliveira, Teresa Cunha-Oliveira, Luciano Saso

**Affiliations:** 1Department of Physiology and Pharmacology “Vittorio Erspamer”, Sapienza University of Rome, 00185 Rome, Italy; 2Section of Hazardous Substances, Environmental Education and Training for the Technical Coordination of Management Activities (DGTEC), Italian Institute for Environmental Protection and Research, 00144 Rome, Italy; 3CNC—Center for Neuroscience and Cell Biology, CIBB—Center for Innovative Biomedicine and Biotechnology, University of Coimbra, 3004-504 Coimbra, Portugal; 4IIIUC—Institute for Interdisciplinary Research, University of Coimbra, 3030-789 Coimbra, Portugal; 5PhD Programme in Experimental Biology and Biomedicine (PDBEB), IIIUC—Institute for Interdisciplinary Research, University of Coimbra, 3030-789 Coimbra, Portugal

**Keywords:** NRF2–KEAP1 pathway, mitochondria, cancer, oxidative stress, ROS, cancer stem cells

## Abstract

The NRF2–KEAP1 system is a fundamental component of the cellular response that controls a great variety of transcriptional targets that are mainly involved in the regulation of redox homeostasis and multiple cytoprotective mechanisms that confer adaptation to the stress conditions. The pleiotropic response orchestrated by NRF2 is particularly relevant in the context of oncogenic activation, wherein this transcription factor acts as a key driver of tumor progression and cancer cells’ resistance to treatment. For this reason, NRF2 has emerged as a promising therapeutic target in cancer cells, stimulating extensive research aimed at the identification of natural, as well as chemical, NRF2 inhibitors. Excitingly, the influence of NRF2 on cancer cells’ biology extends far beyond its mere antioxidant function and rather encompasses a functional crosstalk with the mitochondrial network that can influence crucial aspects of mitochondrial homeostasis, including biogenesis, oxidative phosphorylation, metabolic reprogramming, and mitophagy. In the present review, we summarize the current knowledge of the reciprocal interrelation between NRF2 and mitochondria, with a focus on malignant tumors and cancer stem cells.

## 1. Introduction

The Nuclear factor erythroid 2-like 2 (Nfe2l2/NRF2) is a transcription factor belonging to the Cap’n’collar (CNC) subfamily of basic leucine zipper (bZip) proteins that plays a major role in the cellular stress response against oxidative insults of different origins. By controlling the expression of enzymes involved in redox homeostasis, cell metabolism, and xenobiotics detoxification, NRF2 promotes the survival in normal cells, cancer cells, as well as cancer stem cells (CSCs), and this is a consequence of the adaptive response triggered in response to adverse conditions present in the microenvironment or caused by therapeutic interventions. In the following paragraphs, we briefly describe the mechanisms that regulate NRF2 induction and the transcriptional program controlled by its activation. We also describe the functional interrelation between NRF2 activity and mitochondrial functions, and we discuss recent data on the role of NRF2 in cancer and CSCs, underlining how the regulation of redox homeostasis can have profound impacts on CSCs biology and therapy resistance.

## 2. Part 1: The NRF2–KEAP1 Pathway and the Mechanisms Underlying Its Canonical or Non-Canonical Activation

### NRF2 Activation by KEAP1-Dependent or Independent Mechanisms

NRF2 is a very labile protein and is characterized by an extremely short half-life of approximately 20 min. Its activation is primarily controlled at the level of its protein stability, which is controlled by a number of different interactors that are regarded as canonical or non-canonical binding partners that ultimately influence/modulate the NRF2 degradation by the ubiquitin–proteasome system. A fine regulation exists to ensure constitutive activation of the cytoprotective pathways controlled by NRF2 under basal conditions while its steady-state levels are dramatically increased in response to specific stimuli (e.g., oxidative/electrophilic stress or chemical inducers) which, in turn, also enhance the expression of the NRF2 target genes.

Under physiological conditions, NRF2 levels are controlled by its main regulator and canonical interactor, the Kelch-like ECH-associated protein 1 (KEAP1). This dimeric protein sequesters NRF2 in the cytosol, acting as a substrate adaptor for the CUL3–RBX1 E3 ubiquitin ligase complex, facilitating NRF2 polyubiquitylation and its constant degradation by the 26S proteasome [1,2]. However, in the presence of oxidative stress or electrophilic compounds, redox-sensitive cysteine residues (e.g., Cys151, Cys273, and Cys288) within KEAP1 are modified, causing NRF2 dissociation and its subsequent translocation into the nucleus. Here, NRF2 heterodimerizes with small Maf (sMaf) proteins and other transcription factors, such as SP-1 or c-JUN, and binds to specific sequences known as AREs (antioxidant responsive elements) located within the promoter region of different target genes, inducing their transactivation [3,4] (see Figure 1). These events are generally referred to as canonical activation of NRF2, which depends on KEAP1 and is triggered by its oxidative modification. Further investigations have led to the discovery of additional proteins that are capable of influencing NRF2 stability by disrupting its interaction with KEAP1, a mechanism collectively known as non-canonical activation of NRF2 [5]. A vast majority of these regulators, including p62/SQSTM1 [6,7], WTX [8], DPP3 [9,10], PALB2 [11], and Prothymosin α [12,13], possess specific interaction domains with high affinity for KEAP1 and therefore prevent its recognition by NRF2 [9]. Conversely, other proteins, including p21 [14] and BRCA1 [15], act in a complementary way due to the presence of high-affinity domains for NRF2 interaction, thus competing with KEAP1 for NRF2 binding (see Figure 2). Interestingly, the non-canonical activators of NRF2 are primarily involved in the regulation of disparate cellular functions such as autophagy (p62), protein turnover (DPP3), DNA repair (PALB2), cell cycle, and apoptosis (p21, Prothymosin α), while WTX and BRCA1 are well-known oncosuppressors that control genomic stability, cell-cycle progression, cell migration, and stem-cell pluripotency, thus emphasizing the complexity of NRF2 signaling and its intimate connection with cancer. Among the non-canonical regulators of NRF2 activity, the serine/threonine phosphatase Phosphoglycerate mutase 5 (PGAM5) constitutes a peculiar example of a mitochondrially localized protein that interacts the with KEAP1–NRF2 complex at the level of the external mitochondrial membrane. This specific pathway is described more in detail in Section 3.2.

An alternative pathway for NRF2 degradation, which is instead KEAP1-independent, has been also identified in mouse cells and human cancer cells, and it is intimately connected with PI3K/AKT signaling. Indeed, in the absence of AKT activation, NRF2 is phosphorylated by the glycogen synthase kinase 3 beta (GSK-3β), and this event facilitates its recognition by the β-TrCP–Skp1–CUL1–RBX1 E3 ubiquitin ligase complex, which, in turn, promotes NRF2 ubiquitylation and its subsequent proteasomal degradation. By contrast, when upstream signals engage with the PI3K/AKT pathway, the GSK-3β protein is phosphorylated and, thus, inactivated; this relieves NRF2 from proteasomal degradation and leads to its stabilization [16,17,18] (see Figure 3). It has been proposed that these two axes cooperate to regulate the abundance of NRF2 in a distinct way since GSK3/β-TrCP would finely shape NRF2 levels in response to transient changes in cellular metabolism, while the impairment of KEAP1 is believed to induce a much larger increase in the NRF2 pool as a consequence of the exposure to different environmental stressors [19]. While the description of the detailed mechanisms of NRF2 regulation are beyond the scope of this manuscript, the interested reader is referred to other excellent review articles [5,20].

## 3. Part 2: Emerging Evidence of the Interplay between NRF2 and Mitochondria

### 3.1. NRF2 Regulates Mitochondrial Functions (Energetics, Biogenesis, and Mitophagy)

Mitochondria are cellular organelles of great importance due to their involvement in cellular biology. Some of the mitochondria’s intrinsic characteristics, such as possessing their own DNA and double-membrane boundary, make them stand out from other cellular structures. These characteristics can be explained by the symbiogenesis theory [21], which states that, early during eukaryotic cell evolution, a mitochondrial precursor could exist on its own as a separate entity, eventually becoming integrated into the eukaryotic cell functioning [22,23]. Although mitochondria’s best-known purpose is energy production, supplying most of the cellular energy in the form of Adenosine Triphosphate (ATP) through oxidative phosphorylation (OXPHOS), mitochondria are also essential for other aspects of eukaryotic life, being responsible for numerous critical cell functions, including the generation and regulation of reactive oxygen species [24], calcium (Ca^2+^), hormone, and immune-mediated signaling; regulation of cellular metabolism; cell division, differentiation, and proliferation; autophagic degradation (mitophagy); and regulation of apoptosis through mitochondrial outer membrane permeabilization, namely through the mitochondrial permeability transition pore (mtPTP).

Mitochondria are highly dynamic organelles, moving around the cell through the microtubule network, constantly changing their morphology by fission and fusion processes which are tightly controlled to balance possible metabolic or environmental alterations [22,25]. Furthermore, mitochondria are very communicative with the other cellular organelles which manifest their importance for signaling cascades and normal cell function. Mitochondria have their own genome (mtDNA), which is circular and replicates independently of the host genome. In humans, mtDNA encodes 13 proteins of respiratory complexes I, III, IV, and V; peptides, namely eight mitochondrial-derived peptides (MDPs) which are encoded by short open-reading frames (sORF); and non-coding RNAs [26,27], i.e., 22 for mitochondrial tRNA, and 2 for rRNA [28]. The remaining mitochondrial proteins (~1136) [29] are encoded by the nuclear genome and are imported into the mitochondria [29,30,31]. Since mitochondria participate in almost all aspects of cellular homeostasis, the coordination between both genomes and proper interaction with the rest of the cell is crucial to the maintenance of mitochondrial health and, therefore, the prevention of several human diseases. Pathologies associated with improper mitochondrial function are diverse, ranging from neurodegenerative diseases, such as Alzheimer’s disease, to diabetes and even cancer [32]. Dysfunctional mitochondria present several biochemical and morphological changes, as well as loss of membrane integrity, which can induce inflammatory responses and, consequently, cell death [22,31].

As the NRF2–KEAP1 system is a fundamental component of the cellular response and encompasses multiple cytoprotective mechanisms, the importance of NRF2 in mitochondrial function has been investigated and established. Apart from the aforementioned mechanisms, NRF2 and KEAP1 have also been detected at the outer mitochondrial membrane, bound to the mitochondrial serine/threonine-protein phosphatase (PGAM5), which is a mitochondrial protein with an important role in mitochondrial dynamics and homeostasis that interacts with both NRF2 and KEAP1, composing a ternary complex, NRF2–KEAP1–PGAM5, reinforcing the crucial role in the healthy maintenance of mitochondrial function [33,34].

NRF2 transcriptional activity affects many aspects of mitochondrial physiology and homeostasis [35], mitochondrial energetics [36], mitochondrial biogenesis [37], fatty acid oxidation [38], respiration [39], ATP production [40], membrane potential [41], and redox homeostasis [42], as well as the structural integrity [43], motility, and dynamics [34] of this essential organelle. Target genes of NRF2 with ARE elements, which encode for mitochondria-relevant proteins, include *TXN*, *G6PD*, *GSTA2*, *NQO1*, *GSTP1*, and *HMOX1*, among others [44,45]. These genes belong to the antioxidant system, NADH regeneration, REDOX, ROS detoxification, and iron metabolism categories.

All cellular processes need a constant supply of energy, which can be generated by several metabolic pathways that break down complex macromolecules into intermediate metabolites to generate ATP. Two main pathways are involved in cellular energy metabolism: glycolysis and OXPHOS. Glucose is of extreme importance for cellular metabolism, as it is the main source of energy for most tissues, needed to fuel both aerobic and anaerobic respiration. Through glycolysis, glucose gives origin to two pyruvate molecules. Those can be converted into acetyl CoA, which is used in the tricarboxylic acid cycle (TCA) to generate ATP along with the electron transport chain (ETC). Energy homeostasis is critical for cell fate since metabolic regulation is dependent on the right balance between catabolic (mitochondrial OXPHOS to generate ATP from ADP + Pi and NADH in NAD^+^ oxidation) and anabolic (ATP consumption and NADH generation) pathways. The AMP-activated protein kinase (AMPK) is the major energy sensor in eukaryotic cells, also known as the guardian of metabolism and mitochondrial homeostasis [46], participating in several important processes for mitochondrial health, such as autophagy and mitophagy regulation, mitochondrial dynamics, and transcriptional control, as well as mitochondrial biogenesis. Convergence between AMPK and NRF2 pathways is important for the anti-inflammatory effect of berberine in lipopolysaccharide (LPS)-stimulated macrophages and endotoxin-shocked mice [47]. When AMPK is activated, it leads to a reprogramming of metabolism into increased catabolism and decreased anabolism through the phosphorylation of the mammalian target of rapamycin (mTOR) [46]. Moreover, mTOR can counteract AMPK regarding the control of catabolism and anabolism for cellular homeostasis maintenance [46]. NRF2 can also directly regulate mTOR through activation of the phosphatidylinositol-3-kinase (PI3K) pathway, which is frequently mutated in the most common human cancers [48]. Moreover, metabolic reprogramming with enhanced glutamine dependence in KRAS-mutant lung adenocarcinoma was shown to be promoted through cooperation between the KEAP1/NRF2 system and LKB1, also known as Serine/Threonine Kinase 11 (STK11), which is a tumor suppressor and the major upstream activator of AMPK, establishing a link between energy metabolism and tumor suppression [49]. Energy metabolism deregulation is a well-known characteristic of cancer. Tumor cells adapt to environmental changes, rewiring their metabolism to compensate for high energy requirements. Often, this shifts the metabolic profile in cancer cells, which become more glycolytic, with glucose being fermented into lactate, even in situations of an abundance of oxygen in which OXPHOS would otherwise be favored. This shift in metabolic phenotype, characteristic of several tumors, is known as the Warburg effect [50]. For both physiologic and pathological conditions, mitochondrial biogenesis is initiated in response to energy demand triggered by stress signals. There are several possible explanations for what drives this phenomenon. An early explanation was that it allows for the synthesis of more substrates to be used in the biosynthesis of macromolecules, such as nucleic acid, lipids, and proteins, much needed to allow their rapid proliferation rate, but this hypothesis was contested, as most of the glucose’s carbons are excreted into lactate, with amino acids being a preferred source for this purpose [51]. Other explanations include that it allows for a faster ATP production despite being less efficient than through respiratory pathways [52], or that it is due to molecular crowding [53]. As such, this is an important topic that requires better clarification [50,54].

Mitochondrial biogenesis (Figure 4) is a biological process by which mitochondria self-replicate. This process is complex and closely regulated, requiring tight coordination between the mitochondrial and nuclear transcription factors [55]. PGC-1α, a member of the peroxisome-proliferator-activated receptor gamma family of transcriptional coactivators, is considered to be the master regulator of mitochondrial biogenesis. Together with NRF2, it coactivates nuclear respiratory factor 1 (NRF1) and therefore activates, upon AKT phosphorylation and GSK3β inactivation, the mitochondrial transcription factor A (TFAM), which is required for the maintenance of normal levels of mitochondrial DNA [24]. Thus, mitochondrial biogenesis markers may include mtDNA:nDNA ratio and the expression levels of mitochondria-related genes, such as PGC-1α, TFAM, nuclear respiratory factor 1 (NRF1), GABP (GA-binding protein), and mitochondrial transcription factor B1 (TFB1M). On a side note, GABP is also known as nuclear respiratory factor 2), being commonly confused with the NRF2 (nuclear factor-erythroid-derived 2-like 2) covered in the present review [56]. The NRF2-target gene heme oxygenase-1 (*Ho-1*, also known as *Hmox1*) was found to stimulate mitochondrial biogenesis via NRF2 and Akt in mouse heart [57]. Later, the same workgroup showed that mitochondrial biogenesis is associated with an increase in the expression of two major anti-inflammatory genes, *IL10* and *IL1Ra*, through HO-1/CO and NRF2 redox regulation in mitochondria from human-hepatoma-derived HepG2 cells and in vivo liver cells [58]. A role of NRF2 in mitochondrial respiration was suggested when NRF2 was knocked down in the human colon cancer cell line through blockade of HIF-1α signaling [59]. In addition to mitochondrial gene expression, mitochondrial biogenesis also requires the synthesis of nucleotides and phospholipids. NRF2 promotes genes involved in the Pentose Phosphate Pathway (PPP), De Novo Nucleotide Synthesis and NADPH production [60], purine biosynthesis, and glutamine metabolism by activating PI3K–AKT signaling [60].

Mitochondrial function and structure are notoriously influenced by the phospholipid composition of their membranes. Some must be imported, such as phosphatidylcholine (PC), phosphatidylinositol (PI), phosphatidylserine [30], and phosphatidic acid (PA), or synthesized inside mitochondria, including phosphatidylglycerol (PG), phosphatidylethanolamine (PE), and cardiolipin (CL), which is exclusively located at mitochondria inner membrane [61]. Defects in phospholipid levels are associated with mitochondrial dysfunction and may lead to the development of severe diseases [62]. The interaction between phospholipids and proteins is important, particularly in the inner mitochondrial membrane, where oxidative phosphorylation takes place. Mitochondrial oxidation of palmitic (long-chain) and hexanoic (short-chain) fatty acids is either diminished or enhanced, respectively, if NRF2 is absent or constitutively active in heart and liver mitochondria [38]. An analysis of the oxygen consumption rate revealed altered maximal respiration most probably due to differences in substrate (namely palmitoleic acid and oleic acid) availability [38]. Moreover, for the coenzyme FADH2 in the flavoprotein of mitochondrial complex II, its generation was lower in *Nrf2*-KO murine neurons and embryonic fibroblasts when compared with both WT or constitutively active Nrf2, indicating that Nrf2 impacts cellular bioenergetics both in vivo and in vitro [39].

Mitochondrial homeostasis requires tight coordination between the generation of new mitochondria by mitochondrial biogenesis and the removal of damaged mitochondria by mitophagy, a process whereby dysfunctional mitochondria are engulfed by autophagosomes and delivered to lysosomes to be degraded and recycled [63]. When mitochondrial integrity is lost, these organelles lose osmotic homeostasis, and this can lead to cell death through, for example, induction of the mtPTP. Induction of NRF2 upon administration of isothiocyanate sulforaphane to rats increases the resistance of isolated brain mitochondria to redox-regulated mtPTP opening [43], suggesting a role for the NRF2 pathway in mitochondrial integrity (Figure 4). Selective autophagic adaptor protein sequestosome-1 (SQSTM1/p62) is critical for this process and possesses a functional ARE. Moreover, p62 interacts with the NRF2-binding site on KEAP1, competing with NRF2 for binding, when overproduction or deficiency of p62 occurs in autophagy, resulting in transcriptional activation of NRF2 target genes. Pathological processes associated with the excessive accumulation of p62 were found to result in hyperactivation of NRF2, causing liver injury [63]. Another study showed that persistent activation of NRF2, through the accumulation of phosphorylated p62, contributes to the growth of human hepatocellular carcinomas, indicating that KEAP1–NRF2 and selective autophagic pathways are connected through the phosphorylation of p62 [7]. Somatic mutations in KEAP1 and NRF2 have been identified in patients with different types of cancers, including lung, head and neck, and gallbladder, and these mutations result in the disruption of the KEAP1–NRF2 interaction and persistent activation of NRF2 through binding with ARE regions [64,65,66]. Mitophagy also operates via the mitochondrial serine/threonine-protein kinase PTEN-induced kinase 1 (PINK1)/Parkin pathway targeting mitochondria with low membrane potential (ΔΨm) to autophagosomes. The compound p62-mediated mitophagy inducer (PMI) was found to increase the expression and signaling of the autophagic adaptor molecule p62/SQSTM1 and force mitochondria into autophagy [67]. PMI was shown to induce mitophagy independently of ΔΨm dissipation and of the mitochondrial PINK1/Parkin pathway [67]. Ultrastructural analysis of hepatocytes from Nrf2-KO mice, but not WT, that had been fed with a high-fat diet for 24 weeks showed swollen mitochondria with reduced cristae and disrupted membranes [68]. As such, there is enough evidence to conclude that NRF2 plays a critical role in mitochondrial integrity maintenance under oxidative and inflammatory stresses. Mitophagy is regulated by several points, one being the p62–NRF2–p62 Mitophagy Regulatory Loop, which can be a key target for the treatment of mitochondria-related diseases, including the neurodegenerative ones [69]. In addition, p62 possesses a KEAP1-interacting domain that can lead to the stabilization of NRF2. This KEAP1/NRF2/p62 feedback loop has been shown to be important in several pathologies. For instance, it has a protective effect through oxidative-stress-induced autophagy that may be of interest not only for anticancer strategies [70] but also in other conditions or in the prevention of disease, for example, in the degeneration of the intervertebral disks prevention [71].

Some examples of molecular NRF2 activators which improve mitochondrial function include pharmaceutical compounds such as PMI [67,72], Bardoxylone [73], RTA-408 [74], and dimethyl fumarate [75]. Activators of NRF2 also include dietary compounds, such as curcumin, hesperidin, quercetin, steviol glycosides, and sulforaphane, among others, which can ultimately cause the activation of different proteins and increased levels of intracellular antioxidants. This shows that diet can also be an important asset in therapeutic approaches and a possible source of inspiration for new therapies with high patient compliance, possibly even exploiting this facet with other known effects of those compounds, such as the metabolic remodeling induced by some steviol glycosides [76,77,78,79,80,81,82,83].

### 3.2. Mitochondrial ROS and Mitochondrial Proteins Can Modulate NRF2 Activity

As referred to in the previous section, the correct balance of mitochondrial fission/fusion, turnover (biogenesis/mitophagy/repair, calcium, and ROS homeostasis) is important for the maintenance of healthy mitochondria [84,85]. When this balance is disrupted, mitochondria can become dysfunctional with overproduction of Reactive Redox Species (RRS) and increased nitrosative or oxidative stress, leading to numerous human pathologies [84,86]. NRF2 is a ROS-sensing nuclear factor, and imbalances in their quantity, due to overproduction by mitochondria, lack of ability to remove them, or other factors, can cause changes in NRF2 levels as the cell tries to adapt, altering several cellular functions, including glucose and lipid homeostasis [86]. ROS is an umbrella term that includes several chemically reactive molecules derived from oxygen, containing free (unpaired) electrons, such as the superoxide radical anion (O_2_•^−^), the hydroxyl radical (HO•), and peroxyl radicals (ROO•), as well as nonradical derivatives of molecular oxygen (O_2_), such as hydroperoxides (hydrogen peroxide (H_2_O_2_) and tert-Butyl Hydroperoxide (t-BHP)), hypochlorous acid (HOCl), singlet oxygen O_2_, and peroxynitrite (ONOO^−^) [87,88]. On a side note, the term “ROS” is very generic and often used improperly. One should be careful to avoid the general term and refer to the specific species [89,90,91]. Notwithstanding, ROS are produced in several cell compartments, such as cell membrane, cytoplasm, endoplasmic reticulum, peroxisomes, Golgi apparatus, and, most prominently, mitochondria, which are the major source of ROS, with about 90% of cellular ROS being produced by them [92,93,94]. An increase in mitochondrial biogenesis, with consecutively higher OXPHOS activity, leads to an increase in the production of endogenous ROS from oxidative metabolism due to the activation of respiratory chain components and TCA cycle enzymes [95]. To counterbalance ROS, cells have an efficient antioxidant system composed of enzymatic and non-enzymatic antioxidants to deal with generated ROS. Enzymatic antioxidants include catalase, CAT (found mainly in peroxisomes and less in mitochondria), glutathione peroxidase, GPx (located both in cytoplasm and mitochondria), glutathione reductase, GR (located both in cytoplasm and mitochondria), glutathione-S-transferase, GST (located in the cytosol), NADP oxidase (located in membrane and cytosolic components), peroxiredoxins (various intracellular locations), and superoxide dismutases [96], including three isoforms—Cu, Zn SOD, and SOD1 located in the mitochondrial intermembrane space and cytosol; Mn-SOD and SOD2 located in the mitochondrial matrix; and Cu, Zn SOD ecSOD, and SOD3 located in the extracellular space. Non-enzymatic molecules with antioxidant function are low-molecular-mass compounds, some are hydrophilic and predominantly present in the cytosol or cytoplasmic matrix (vitamins A, C, and E; flavonoids; uric acid; and albumin), and others are lipophilic and present in cell membranes (carotenoids, ubiquinol, and α-tocopherol). Different antioxidant enzymes may be correlated. For example, there is a close connection between CAT, GPx, and SOD, since H_2_O_2_—the product of SOD reaction—is the substrate for both CAT and GPx [97]. In addition to mitochondrial biogenesis and mitochondrial homeostasis, NRF2 also has a well-recognized role in the maintenance of cellular redox homeostasis by controlling ROS production through regulating the biosynthesis, utilization, and regeneration of glutathione (GSH), thioredoxin, and NADPH [98]. NRF2 activation stimulates mitochondrial antioxidant enzymes GR, GPx, thioredoxin 2, peroxiredoxin 3 (Prdx3), peroxiredoxin 5 (Prdx5), and SOD2, among others, to counteract the increased ROS production in response to oxidizing conditions (Figure 5). Nevertheless, the exact mechanisms are not fully understood [99,100,101,102,103]. NRF2 can influence mitochondrial activity as direct or secondary action and in different ways, namely by rewiring cellular metabolism to account for the needs of metabolic intermediates such as Fructose 6-Phosphate (F6P) and Glyceraldehyde 3-Phosphate (G3P) between glycolysis and PPP [104]. On the other side, selective stimulation of mitochondrial ROS by MitoPQ did not lead to an activation of the NRF2 pathway, in contrast with the effects evoked by MitoCDNB-related impairment of the pool of mitochondrial GSH and inhibition of the mitochondrial thioredoxin system, which activated NRF2 [105].

There are reciprocal regulatory loops between NRF2 and mitochondrial proteins, including PGC1-α, DJ-1, PGAM5, frataxin, p62, and others [106], thus further highlighting the interdependence between NRF2 and this organelle. PGC1-α, as referred to in the previous section, is a major regulator of mitochondrial biogenesis and is also involved in the antioxidant defense, including by modulating the *SOD2* gene transcription, as well as protein levels [24]. The cytoprotective effects of PGC1-α have been linked with an upregulation of NRF2, impaired with an NRF2-specific knockout, in a mechanism mediated by GSK3β inactivation through activated p38 [107]. In turn, targeting NRF2 with siRNA decreases the levels of PGC1-α [108].

The increased levels of ROS that are often present in cancer can lead to retrograde signaling through the JNK–PGC1-α pathway with increased complex II phosphorylation and enhancement of mitochondrial biogenesis [109,110]. As described above, PGAM5 is of great importance in several facets of mitochondrial function. It interacts with both NRF2 and KEAP1, linking them with mitochondria, with knockdowns of either PGAM5 or KEAP1, leading to an increase in the levels of NRF2, thus hinting at the repression of NRF2-dependent gene expression by PGAM5 [33]. There are cases in which inhibition of PGAM5 can be of therapeutic value, not only for cancer but also for other pathologies, through the use of LFHP-1c, a new PGAM5 inhibitor, which, in this case, allowed for enhanced activation of NRF2, with promising results for brain ischemic stroke by counteracting ischemia-induced Blood–Brain Barrier (BBB) disruption [111]. In another instance of a regulatory loop, NRF2 influences and is influenced by the levels of mitochondrial ROS. An example of this is the management of the levels of mitochondrial GSH, whose biosynthesis can be regulated by NRF2, and which, upon reaction with some reactive molecules such as H_2_O_2_ and organic hydroperoxides, leads to the formation of glutathione disulfide (GSSG). GSSG can be converted back into GSH by the action of GSH reductase, whose production is also regulated by NRF2. As such, through this process, NRF2 participates in the maintenance of the mitochondrial pool of GSH [44,112].

NRF2 is also involved in the regulation of several metallic ions’ redox activity, namely iron, as it plays a key role in iron homeostasis [113]. Iron oxidation is closely interconnected with oxygen transport, consumption, and even the production of ROS [114]. NRF2-mediated upregulation of iron-related proteins, such as heme-oxygenase for transport of oxygen and ferritin for iron storage, can be associated with several pathologies, including cancer [114]. Iron is also present in the event known as ferroptosis, a type of non-apoptotic regulated cell death, driven by iron-dependent lipid peroxidation, in a process that is interconnected with mitochondrial function [115]. In the case of mitochondrial ferritin, it may have protective effects in the inhibition of ferroptosis [116]. CISD1, an iron-containing outer mitochondrial membrane protein, was shown to interact with NRF2-mediated transcription to prevent mitochondrial injury during ferroptosis [117]. The pathway p62–NRF2 was shown to be involved in the resistance of cancer cells to ferroptosis under GPX4 inhibition [118]. Additionally, inhibition of the ETC or the TCA cycle has been linked with a reduction in ferroptosis, while also having a role in cysteine-deprivation-induced ferroptosis, thus suggesting that this necrotic event can be involved in tumor suppression [119].

Frataxin is another mitochondrial protein that has been identified as a regulator of ferroptosis, and its suppression has been associated with a decrease in the NRF2–KEAP1 axis [120]. Despite all of this information, the exact mechanisms underlying the role of mitochondria in ferroptosis are yet to be fully elucidated. However, targeting NRF2 and ferroptosis shows great promise in new treatments and approaches to deal with several pathologies, including cancer [118]. Several publications have documented a close relation between NRF2 and mitochondrial ROS homeostasis. Upregulation of peroxiredoxins 3 and 5 by quercetin in an in vitro model of trabecular meshwork was shown to be dependent on the NRF2/NRF1 pathway, which protects against ocular-disease-induced oxidative stress [121]. NRF2 showed a neuroprotective response upon angiotensin-II-induced oxidative stress in neurons in a model of Parkinson’s disease, both in vivo and in vitro [122]. MitoQ, a mitochondria-targeted antioxidant, protected intestinal epithelial cells (IECs) from mtDNA damage induced by ischemia/reperfusion (I/R) through the activation of the NRF2 signaling pathway [123]. Another mitochondria-targeted antioxidant molecule, AntiOxCIN4, showed a significant cytoprotective effect in human hepatoma-derived (HepG2) cells through the NRF2–p62–KEAP1 axis [124]. NRF2 deficiency found in an experimental autoimmune encephalomyelitis in vitro model was associated with reduced synthesis of GSH, leading to the production of pro-inflammatory mediators in this disease [99].

Keeping in mind all of this information, current knowledge points toward an interplay between NRF2 and the mitochondrial network which can occur through a direct interaction of this transcription factor with some mitochondrial proteins or be the result of a fine-tune shaping of mitochondria-derived ROS. This functional crosslink has an emerging potential to be the cornerstone for novel therapeutic approaches to tackle a wide array of human pathologies, including cancer.

## 4. Part 3: The Crosstalk between NRF2 Signaling and Mitochondria in Cancer and Cancer Stem Cells

### 4.1. NRF2 Enhances the Antioxidant Ability of Cancer Cells to Mediate Survival and Adaptation

Multiple lines of evidence indicate that malignant cells generate large amounts of intracellular ROS compared to normal cells, as a consequence of genetic, epigenetic, and metabolic disturbances that support cell proliferation and drive malignant progression. To overcome this augmented ROS production and prevent cytotoxic effects, cancer cells have evolved adaptive mechanisms which include reprogramming of specific metabolic pathways, the increased expression of ROS-scavenging enzymes, DNA repair systems, regulators of protein thiols, and proteins involved in the storage of redox-active metals. Taken together, these systems ultimately lead to an augmented antioxidant capacity of cancer cells. This phenotypic alteration, which has been widely observed in a great variety of solid and hematologic tumors, is now recognized as an important hallmark of cancer cells [125]. Importantly, the activation of NRF2 signaling can also have profound impacts on the biology of cancer stem cells (CSCs), influencing their tumorigenicity, stemness, and survival. The role of NRF2 in CSCs is discussed in greater detail in the following sections.

### 4.2. NRF2 Controls the Redox Balance of CSC to Mediate Stemness, Tumorigenicity, and Therapy Resistance

It is increasingly recognized that malignant tumors are actually heterogeneous entities constituted by multiple cell populations, including cancer cells, immune cells, stromal cells, and stem cells. CSCs represent a small subpopulation of malignant cells that have been initially identified in acute myeloid leukemia (AML) [126]. Later work has shown that CSCs are also present in solid tumors such as brain, breast, colon, head and neck, liver, lung, pancreas, and prostate cancers [127,128,129,130,131,132,133,134]. CSCs possess intrinsic abilities of self-renewal, differentiation, tumor initiation, and malignant progression, which make them largely responsible for tumor resistance and recurrence after therapy [135]. While malignant cells are characterized by genetic epigenetic and metabolic changes that cause disturbances in the intracellular redox balance toward pro-oxidizing conditions, whether similar alterations are also present in CSCs and how these subpopulations regulate ROS production/elimination is poorly understood [127]. However, it is known that both malignant cells and CSCs strongly rely on NRF2 signaling to respectively prevent cell demise and to maintain their self-renewal capacity, leading to increased cell survival and therapy resistance [106,136,137]. In this regard, evidence indicates that different subpopulations of CSCs maintain lower intracellular ROS levels compared to their non-stem counterparts through a variety of different mechanisms, including the enhanced expression of intrinsic regulators of redox balance, antioxidant enzymes, and the modulation of the energetic metabolism. Regarding the former cases, a seminal paper from Diehn et al. investigated the correlation between intracellular ROS levels and radioresistance in the context of breast cancer. Here the authors analyzed different subpopulations from surgically resected human breast tumors to identify a CSC subpopulation CD44^+^/CD24^−^ that was characterized by lower intracellular ROS levels compared to the non-tumorigenic counterpart. Transcriptional profiling by microarray and Gene Set Enrichment analysis revealed that the CSC exhibited an increased expression of *GCLM* and *GSS*, two enzymes involved in GSH synthesis; and *FOXO1*, a transcription factor that controls the expression of antioxidant genes. Similar observations were also seen in mammary tumors from a mouse model of breast cancer [138]. Importantly, treatment with buthionine sulfoximine (BSO), a pharmacologic inhibitor of GSH synthesis, enhanced the radiosensitivity of the Thy1^+^CD24^+^Lin^−^ CSC-enriched cells isolated from MMTV-Wnt-1 breast tumors by increasing their intracellular ROS levels [138]. Other work from Herault et al. was focused on leukemia stem cells (LSCs) of human origin derived from human AML. Here the authors analyzed two leukemia clones characterized by a different abundance of LSC subpopulations, namely FLA2 (high LSC frequency) and FLB1 (low LSC frequency). A comparative mRNA profiling by microarray and the promoter methylation analysis by sodium bisulfite showed hypomethylation of the *GPX3* (glutathione peroxidase 3) promoter region in FLA2 cells compared to the FLB1 counterpart, and this resulted in the higher expression of this antioxidant enzyme at the mRNA and protein level and was associated with lower intracellular ROS. Genetic silencing of *GPX3* by shRNA strongly decreased the proportion of FLA2 cells contributing to fulminant leukemia after their transplantation into mice recipients [139]. Other work has provided evidence that specific CSC markers can also participate in the regulation of redox balance by indirectly influencing antioxidant systems. For example, Ishimoto et al. analyzed three gastric cancer lines (MKN28, AGS, and KATOIII) and two colorectal cancer lines (HT29 and HCT116) to reveal that the expression of the CSC marker CD44 and its variant (CD44v) was correlated with an increased antioxidant efficiency in response to H_2_O_2_ treatment that was abrogated by CD44 depletion with RNAi. Transcriptional analysis by qPCR on CD44^+^ or CD44^−^ isolated from the gastric tumors of 30-week-old transgenic mice (*K19-Wnt1/C2mE* or *Gan*), confirmed that the relative mRNA abundance of H_2_O_2_ scavenging enzymes such as glutathione peroxidase (*Gpx1* and *Gpx2*) and peroxiredoxin isoforms (*Prdx1* and *Prdx4*) was significantly higher in CD44^+^ than in CD44^−^ tumor cells. Surprisingly, however, the expression or the protein content of these enzymes was not affected by genetic silencing of CD44 in cultured cells, but this led to a significant drop in the GSH levels in HCT116 cells. Intriguingly, further investigations revealed that a variant of CD44 (CD44v), but not CD44s, promoted GSH synthesis through the interaction and subsequent stabilization of the xCT subunit of the antiporter known as the xc^−^ system, which mediates the uptake of cystine, a precursor in the biosynthesis of GSH. Moreover, CD44 genetic ablation in transgenic *Gan* mice induced growth arrest in the immature and proliferative tumor cells by inducing cell differentiation as a consequence of increased activation of the p38 MAPK and its downstream component p21CIP1/WAF1 [140]. In a subsequent work, the same group investigated the functional interrelation between CD44v and xCT in human head and neck squamous cell carcinomas (HNSCC). Here, the authors found that the expression of the CD44 variant was essential to support xCT activity and maintain the intracellular redox balance by increasing the intracellular GSH levels to keep low steady-state ROS levels. Importantly, the chemical inhibition of xCT by sulfasalazine or its genetic silencing impaired cell proliferation and selectively induced apoptosis in undifferentiated CSCs with high CD44v expression in vitro, while sulfasalazine enhanced cisplatin toxicity, which is in part mediated by ROS, in differentiated-type tumors formed by HSC-2 cells in vivo [96]. In a different context, Haraguchi et al. took advantage of human liver cancer cells, HuH7, and PLC/PRF/5, as well as HCC clinical samples, to identify a dormant subpopulation of slow-cycling cells with enriched expression of the CD13 (Aminopeptidase N) marker. These cells were found to initiate tumorigenesis when transplanted in NOD/SCID mice and to possess a therapy-resistant phenotype, regardless the concurrent expression of other stemness markers, such as CD133 or CD90. Of note, measurements with 2′,7′-dichlorofluorescein diacetate (DCF-DA) revealed that the CD13^+^/CD133^+^ and the CD13^+^/CD90^−^ fractions of the HCC cells contained lower basal levels of cytosolic and mitochondrial ROS than the CD13^−^/CD133^+^ and CD13^−^/CD90^+^ fractions and were also characterized by an increased ROS scavenging ability in response to exogenous H_2_O_2_ administration. Importantly, treatment with CD-13 neutralizing antibodies or the anticancer drug ubenimex increased the intracellular ROS to levels comparable with those seen in CD13^−^ cells. A further analysis revealed that, in the CD13^+^/CD90^−^ fraction, the modulatory subunit of the glutamate-cysteine ligase (GCLM), an enzyme catalyzing the rate-limiting step of GSH synthesis, was overexpressed compared to the CD13^+^/CD90^+^, CD13^−^/CD90^+^, and CD13^−^/CD90^−^ fractions of PLC/PRF/5 and primary HCC cells. Of note, the inhibition of CD13 function enhanced the sensitivity of these previously resistant subpopulations to oxidative stress induced by irradiation and chemotherapy in HCC cells, leading also to extensive DNA damage in mice xenografts and impaired tumorigenicity of CD13^+^ enriched subpopulations [141]. Later evidence from the same group showed that the increased expression of CD13 in response to TGF-beta-induced epithelial-to-mesenchymal transition (EMT) promoted redox adaptation and cell survival in liver CSC by increasing their ROS scavenging capacity [142]. Moreover, it has been reported that the upregulation of ROS scavenging enzymes in CSC can increase cells’ adaptation to hypoxic conditions. In this respect, Peng et al. showed that CSC isolated from human pancreatic cancer cells Panc-1 expressed higher levels of cytosolic (GPX1 and SOD1) and mitochondrial (SOD2) antioxidant enzymes than their parental counterpart, thus resulting in an increased survival rate. Higher GPX4 expression in Panc1-CSCs was also observed under basal conditions and was not affected by hypoxia, suggesting its implication in more physiological processes. *GPX4* knockdown by siRNA impaired Panc1-CSC stemness properties under normal conditions and decreased their resistance to oxidative stress induced by erastin or H_2_O_2_ administration, resulting in a marked increase in the intracellular ROS levels and in the rate of cell death [143]. Moreover, in a recent study, Won Son et al. investigated the role of peroxiredoxin 2 (PRX2) in the modulation of ROS levels and CSC stemness in different human hepatocellular cancer (HCC) cell lines. The authors showed that oxidative stress induced by H_2_O_2_ treatment impaired sphere formation and the expression of stem cell markers in CSC populations derived from Huh7 (EpCAM^+^/CD133^+^) and SK-HEP1 (EpCAM^+^/CD90^+^) cells. *PRX2* knockdown with siRNA and the overexpression of a peroxidase inactive mutant (PRX2C51/172S) were able to phenocopy these alterations, leading to a marked increase in the intracellular ROS levels, while opposite changes were induced by the overexpression of a WT-PRX2 form and a H_2_O_2_-resistant mutant (PRX2ΔYF), which enhanced CSC stemness even under oxidative stress conditions. Therefore, the authors concluded that PRX2 function played a key role in linking ROS scavenging to CSC stem properties.

It has to be noted that, while the status of NRF2 was not directly investigated in these studies, GCLM, GPX2, GPX4, PRX1, and the xCT system are well-recognized targets of NRF2, suggesting that the increased expression of redox regulators in CSCs might be caused by NRF2 overactivation, similarly to what is observed in non-stem cancer cells [42]. This is consistent with a more recent study from the group of Surh wherein the authors investigated the molecular mechanisms linking redox status and stemness in breast CSCs. Here, from the analysis of CD24^low^/CD44^high^ breast cancer stem-like cells isolated from MCF-7 mammospheres, it was determined that the increased activation of NRF2 was responsible for GCLC overexpression and increased GSH biosynthesis, leading to low intracellular ROS accumulation. This promoted the nuclear localization of FOXO3 and its binding to the *Bmi-1* promoter, which, in turn, enhanced the self-renewal activity and tumorigenic potential of breast CSCs in vitro and in vivo [144].

The modulation of cellular metabolism is another mechanism through which CSCs can potentially regulate their redox homeostasis. Similar to what has been reported for parental cancer cells, evidence indicates that CSCs also possess a remarkable metabolic plasticity which allows them to switch from mitochondrial metabolism/OXPHOS to glycolysis under specific conditions. In this regard, while a direct role of NRF2 in rewiring key metabolic pathways has been extensively demonstrated in cancer cells of different origin [60,145], whether this also occurs in CSCs still has to be elucidated. However, evidence indicates that, in CSCs, NRF2 acts at the interface of redox and metabolic regulation by controlling the intracellular ROS levels, which, in turn, can influence their stemness properties and induce therapy resistance. In this regard, Dong et al. investigated the metabolic profile of an aggressive and therapy-resistant subtype of breast cancer known as BLBC (basal-like breast cancer), which expressed several EMT markers and CSC-like features. By a comparative gene expression microarray analysis, the authors found that the content of the enzyme fructose-1,6-biphosphatase (FBP1), which catalyzes the rate-limiting step in gluconeogenesis, was high in luminal subtypes and almost negligible in BLBC cells due to hypermethylation of the *FBP1* promoter caused by the SNAIL–G9a–DNMT1 complex. Mechanistic investigations were conducted on established BLBC clones with FBP1 reconstituted expression or luminal cells with *FBP1* knockdown by shRNA. It was shown that the absence/genetic loss of *FBP1* induced a metabolic reprogramming, which substantially suppressed the mitochondrial metabolism by decreasing the oxygen consumption and the production of ROS through the inhibition of the ETC-complex-I activity. As a consequence, the glycolytic flux was significantly enhanced by PKM2 activation, and this also resulted in an increased NADPH/NADP^+^ ratio, suggesting that part of the glycolytic intermediates was shunted to the PPP. These metabolic changes were paralleled by increased CSC properties and tumorigenicity both in soft agar assay and in SCID mice transplants. By contrast, *FBP1* expression in reconstituted BLBC cells and in sh-scrambled luminal cell lines led to increased OXPHOS and ROS production, which was associated with the suppression of tumors sphere formation and the decreased expression of CSC markers. In addition, *FBP1* expression in BLBC cells inhibited tumorigenicity in vitro and suppressed tumor formation in vivo, indicating that the loss of FBP1 is a critical oncogenic alteration in BLBC [146]. While a mechanistic link between NRF2 expression and FBP1 content was not demonstrated by the authors, it is conceivable that NRF2 might indirectly attenuate *FBP1* expression, as is consistent with an earlier study from the group of Yamamoto showing that genetic *Keap1* deletion in mice resulted in a significant downmodulation of a bunch of gluconeogenesis-related genes, including *Fbp1* [147]. As already mentioned, the metabolic plasticity of CSCs allows them to rapidly switch from OXPHOS to glycolysis, presumably depending on the activation of specific oncogenic pathways and how these pathways interact with the conditions existing in the microenvironment. In both cases, CSCs adopt specific mechanisms to limit ROS overproduction. While diverting metabolic intermediates from the glycolytic flux into the PPP is a useful strategy to increase the production of reducing equivalents (i.e., NADPH) that can support NADPH-linked antioxidant systems such as GPx/GR and PRX/TRX/TRXR, the upregulation of antioxidant enzymes can compensate the mitochondrial ROS production when the energetic metabolism is driven by OXPHOS. This is illustrated by a very recent work from Song et al. wherein the association between mitochondrial metabolism and redox systems was investigated in colon CSCs isolated from cell lines with different metabolic profiles reflecting glycolysis (HCT116 and SW480) or OXPHOS (HT29 and SNU-C5) dependence [148]. Regardless of the initial metabolic route, CSCs isolated from these cell lines were characterized by a preferential use of the mitochondrial metabolism to generate ATP, and this was accompanied by an increased content of intracellular ROS, compared to non-CSCs while the more differentiated bulk tumor cells gradually switched to the use of glycolysis. In CSCs, the production of ATP via OXPHOS led to an increased ROS accumulation, which, in turn, was accompanied by an increased expression of sulfiredoxin (Srx) and peroxiredoxin isoforms (Prx1,2,3) at the mRNA and protein levels, a trend that was also confirmed by qPCR analysis of CD133+ CSCs isolated from samples of colon cancer patients. *SRX* depletion through the CRISPR/Cas9 system or siRNA interference sensitized CSCs to different forms of cell death and decreased the stability of PRX1, PRX2, and PRX3 proteins, leading to mitochondrial dysfunction and increased ROS accumulation. Additionally, the lack of *SRX* expression strongly impaired tumor growth and metastasis formation in xenografts and ortho-xenografts in vivo. Mechanistic investigations revealed that NRF2 and FoxM1 expression was increased in CSCs isolated from colon cancer tissues and cell lines compared to non-CSCs counterparts. Moreover, *NRF2* knockdown with siRNA resulted in lower CD133 and *SRX* expression and impaired mitochondrial function, leading to increased superoxide production, as indicated by Mito-SOX staining. The authors concluded that the NRF2 and FOXM1 axis enhanced the mitochondrial OXPHOS and upregulated *SRX*, which, in turn, enhanced peroxiredoxins’ stability to maintain the stemness, tumorigenicity and survival of CSCs through ROS disposal [148]. In summary, these data indicate that the maintenance of an appropriate intracellular redox balance is of crucial importance for CSCs biology, and this is a common feature with more differentiated cancer cells that are part of the tumor mass. It is generally assumed that CSCs have a lower intracellular ROS content than non-CSCs, and this is consistent with what is observed for normal stem cells and their non-stem counterpart. By keeping low intracellular ROS levels through the enhanced activity of multiple antioxidant systems, CSCs control their stemness properties and, therefore, their differentiation into mature cancer cells. Additionally, the increased expression of ROS scavenging enzymes renders CSCs more refractory to oxidative stress, and this often results in the development of therapy resistance. It is currently unknown to what extent cancer cells and CSCs differ in their intracellular ROS content and in their respective antioxidant capacity. It might be also difficult to reconcile the apparently contradictory evidence that low ROS levels in CSCs promote tumorigenesis and high ROS levels drive malignant progression in non-CSC. However, it is important to note that both of these populations are part of a heterogeneous disease and are characterized by some overlapping but also peculiar alterations in key signaling pathways that might confer a distinct sensitivity to ROS molecules and a certain variability in redox signaling events underlying specific biological changes. In this respect, CSC and non-CSC are similar yet distinct subpopulations wherein overlapping but also different signaling and metabolic pathways are activated which, in turn, can contribute to generate a similar yet not identic redox profiles. It is also conceivable that the different redox status of these populations would influence the relative function that CSC and non-CSC exert during the different stages of the disease. For example, some cell populations with low ROS content (as in the case of CSCs) that are required to ensure rapid proliferation and maintain self-renewal will need to reside in a specific niche and contribute to tumor initiation and clonal expansion of the tumor mass. On the other hand, some subpopulations with higher ROS content (as in the case of normal cancer cells) might be involved in tumor progression and metastatic dissemination that typically occur at later stages of the disease. If we consider cancer to be a pathology wherein heterogeneous cell populations with variable redox profiles functionally interact across multiple phases of tumor growth and dissemination, the apparent paradox related to high vs. low ROS levels can actually be seen as additional evidence and an expected consequence of cancer’s complexity. However, further investigation will be required to understand whether the differences in the intracellular ROS content of CSC and non-CSC derive from the magnitude of ROS generation or the efficiency of ROS scavenging systems and what are the signaling events associated with these redox changes. A schematic illustration of CSCs’ specific features, which are characterized by a gradual variation between two extremes potentially reflecting a different extent of NRF2 content/activity, is shown in Figure 6.

## 5. Conclusions and Future Perspectives

Compelling evidence indicates that NRF2 plays a key role in the biology of cancer cells and CSCs, lying at the intersection of redox homeostasis and cellular metabolism. From this perspective, mitochondria represent the contact point between redox events and metabolism of malignant cells since these organelles are the main source for cell energy production but also for endogenous ROS generation. Beyond this, the mitochondrial network is also a crucial signaling platform that can participate in the regulation of cell stemness, cell motility, cell death, and therapy resistance. It is now increasingly recognized that a mutual interaction exists between NRF2 and the mitochondrial network, by virtue of which this transcription factor can influence mitochondria biogenesis, dynamics, turnover, and function. For this reason, NRF2 has emerged as a promising therapeutic target in cancer. However, while some aspects of this functional crosstalk have been investigated to a certain extent in cancer cells, this research field is still in its infancy in regard to CSCs. For example, in terms of energetic metabolism, CSCs can rapidly switch from OXPHOS and glycolysis, depending on the specific conditions, and this also promotes cell adaptation and survival. In this context, NRF2 seems to mainly act downstream and to modulate the expression of redox regulators to shape the intracellular ROS content, while a more direct role of this transcription factor in CSCs metabolic reprogramming, such as the rerouting of glucose and glutamine into the PPP as for normal cancer cells, still needs to be demonstrated. Nevertheless, while further investigations will be necessary to assess whether and to which extent NRF2 can control key metabolic pathways of CSCs, increasing evidence strongly suggests that the reliance on NRF2 signaling can represent a vulnerability of CSCs that can be therapeutically targeted. In line with this of principle, the inhibition of NRF2 in cancer and CSCs characterized by overactivation of its downstream signaling would be beneficial to potentiate the efficacy of therapeutic strategies targeting mitochondrial function and dynamics. Therefore, it is foreseen that this area of interest will rapidly expand in the near future and that a better understanding of NRF2 role in the control of mitochondrial function in CSCs and non-CSCs will pave the way to novel and more effective anticancer strategies.

## Figures and Tables

**Figure 1 cells-11-02401-f001:**
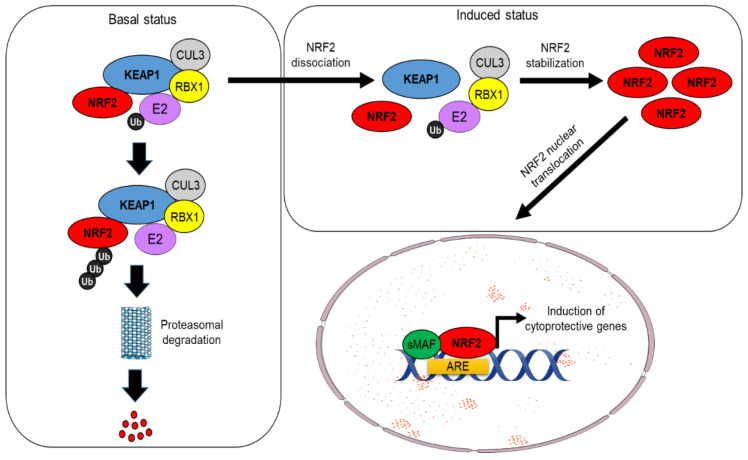
Canonical pathway of KEAP1-dependent NRF2 degradation. Under unstimulated conditions, the interaction between KEAP1 and NRF2 favors its ubiquitination by the CUL3 ubiquitin E3 ligase complex and promotes NRF2 proteasomal degradation in the cytosol. In presence of ROS and other electrophiles, KEAP1 is modified and NRF2 escapes from degradation, enters into the nucleus, and transactivates a large number of cytoprotective genes after heterodimerization with members of the small Maf proteins.

**Figure 2 cells-11-02401-f002:**
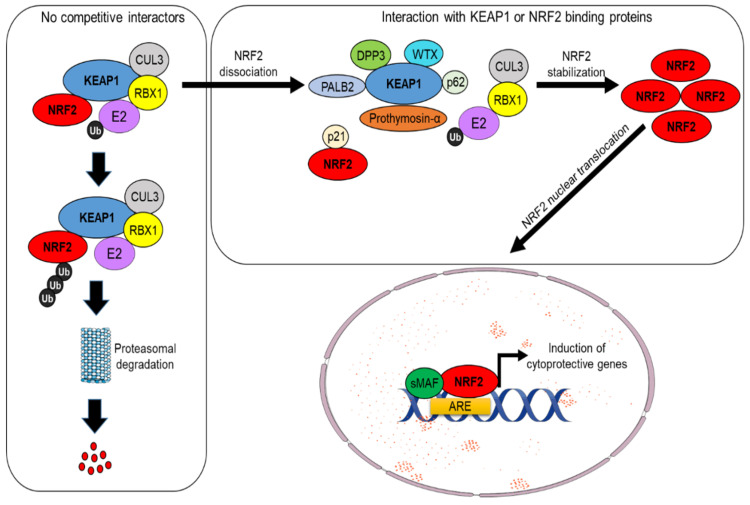
Non-canonical pathway of NRF2 degradation. In the absence of specific interaction partners, NRF2 ubiquitination occurs through the CUL3 ubiquitin E3 ligase complex, which promotes its proteasomal degradation. In the presence of KEAP1 binding proteins that antagonize NRF2 binding, or in the presence of NRF2 interactors that compete for KEAP1 binding, NRF2 escapes from degradation, enters into the nucleus, forms heterodimers with members of the small Maf proteins, and activates the transcription of many cytoprotective genes.

**Figure 3 cells-11-02401-f003:**
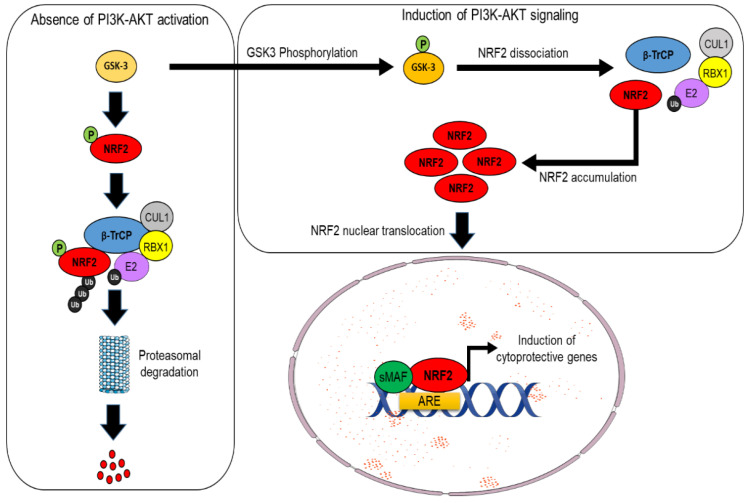
Non-canonical pathway of β–TrCP dependent (KEAP1-independent) NRF2 degradation. In absence of PI3K–AKT upstream signaling, the serine threonine kinase GSK-3 phosphorylates NRF2, and this post-translational modification facilitates its recognition by the CUL1 ubiquitin E3 ligase complex which in turn primes NRF2 for proteasomal degradation. The induction of the PI3K-AKT pathway phosphorylates and inactivates GSK-3, which cannot phosphorylate NRF2. This prevents NRF2 recognition by the CUL1 ubiquitin E3 ligase complex and promotes its accumulation. NRF2 then enters into the nucleus, heterodimerizes with members of the small Maf proteins, and induces the expression of target cytoprotective genes.

**Figure 4 cells-11-02401-f004:**
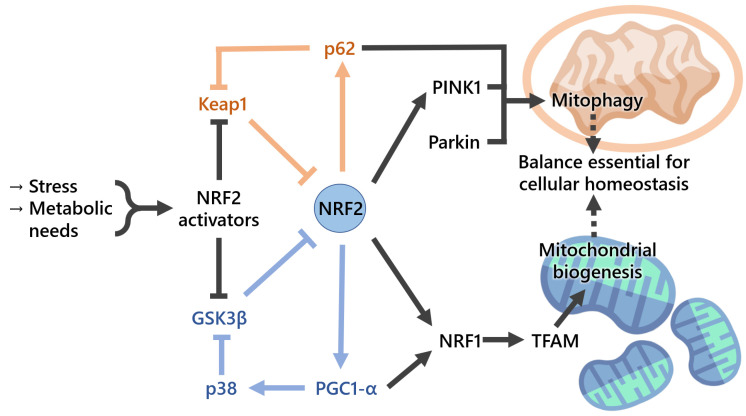
NRF2 plays an important role in mitochondrial and cellular homeostasis. Simplified representation of mitophagy and mitochondrial biogenesis pathways mediated by p62 and PGC1-α, respectively, upon activation by NRF2, evidencing the regulatory loop involving p62, KEAP1, and NRF2 (orange), correlated with mitophagy, and the regulatory loop involving PGC1-α, p38, GSK3β, and NRF2 (blue), associated with mitochondrial biogenesis. NRF2 activators include native proteins, such as AKT, but also xenobiotics, such as RTA-408 or sulforaphane, with the potential to be used in therapeutic strategies.

**Figure 5 cells-11-02401-f005:**
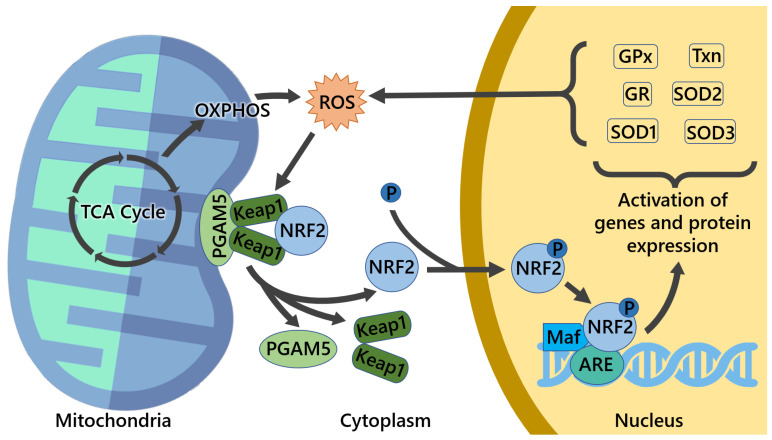
Modulation of NRF2 by mitochondrial ROS leads to the activation of ARE-associated genes and the expression of antioxidant proteins such as GPx (glutathione peroxidase), GR (glutathione reductase), TXn (thioredoxin), SOD1, SOD2, and SOD3 (superoxide dismutase 1, 2, and 3, respectively). When redox imbalance occurs, free radicals are generated, inducing NRF2 release from KEAP1, which, upon proteasomal degradation, translocates to the nucleus and binds to ARE to start the transcription of antioxidant enzymes to, in turn, counteract the oxidative stress from mitochondria and/or the whole cell. PGAM5 is a mitochondrial protein with an important role in mitochondrial dynamics and homeostasis that interacts with both NRF2 and KEAP1, composing a ternary complex, NRF2–KEAP1–PGAM5, reinforcing the crucial role in the health maintenance of mitochondrial function and homeostasis.

**Figure 6 cells-11-02401-f006:**
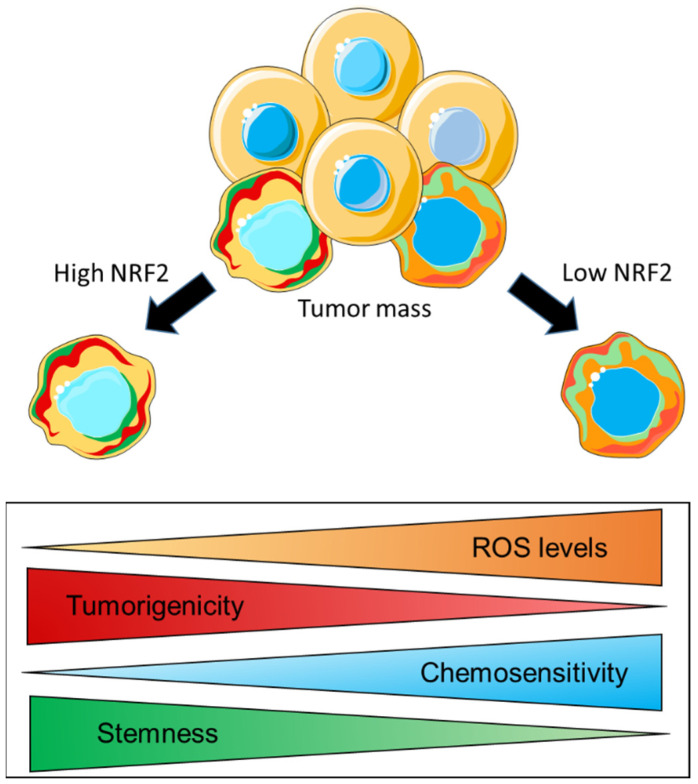
NRF2 regulates crucial aspects of CSCs’ biology by controlling their redox homeostasis. As shown in this schematic representation, within the tumor mass, normal cancer cells and CSCs coexist. Among them, two different phenotypes are observed, most likely reflecting the extent of NRF2 activation. The CSC on the bottom left is characterized by high NRF2 content, low intracellular ROS levels, and high stemness and tumorigenicity but low chemosensitivity. On the opposite side of this spectrum, the CSC on the bottom right has low NRF2 content, low tumorigenicity, and a high content of intracellular ROS, resulting in high differentiation and high chemosensitivity. The figure was created starting from freely available cartoons (https://smart.servier.com/smart_image/in-situ-cancer/, accessed on 27 June 2022).

## Data Availability

Not applicable.

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
