# Peer review of "NRF2 and Mitochondrial Function in Cancer and Cancer Stem Cells"

_cells, 2022, doi:10.3390/cells11152401_

Round 1

Reviewer 1 Report

This is a clearly-written review summarizing the roles of NRF2 and related signaling pathways that regulate and interact with mitochondrial function with a focus on malignancy and cancer stem cells. I recommend this manuscript being accepted with minor revisions as follows.

The authors make the points that: 1) malignant cells are pro-oxidation with high ROS and ROS-scavenging capacity (e.g., line 480-481); 2) cancer stem cells have low ROS with high tumorigenicity (e.g. Figure 6). It is also known the high ROS drive malignancy (e.g., Geou-Yarh Liou and Peter Storz, Reactive oxygen species in cancer, Free Radic Res. 2010 May; 44(5). PMCID: PMC3880197). It may be difficult for readers to reconcile these two apparently opposite features existing in cancer cells. Could the authors add some discussion about this apparent contradiction and possible ways of reconciliation?

This article talks about NRF1 as nuclear respiratory factor 1 and another NRF2 as nuclear respiratory factor 2, which is not the NRF2 covered by the review. The paper does not distinguish them with distinct notations, which can be confusing. For example,

Line 239-242: “NRF1 and NRF2 (the latter also known as GA-binding protein- GABP to avoid confusion with the nuclear factor, erythroid 2-like 2), and mitochondrial transcription factor B1 (TFB1M). The NRF2-target gene heme oxygenase-1 (Ho-1 also known as Hmox1) was found to stimulate mitochondrial biogenesis via NRF2 and Akt in mouse heart”. It is not clear which NRF2 the 2nd NRF2 covered in this paragraph is.

This reference might be helpful. “Punctum on two different transcription factors regulated by PGC-1α: Nuclear factor erythroid-derived 2-like 2 and nuclear respiratory factor 2”

Line 93: “Akt signaling can promote the phosphorylation of NRF2 by the glycogen synthase kinase 3 beta (GSK-3β) …” This statement is confusing. It appears to suggest increased Akt signaling activity promotes NRF2 phosphorylation and thus its degradation, which is contradictory with that, without PI3K/Akt activation, GSK-3 will phosphorylates NRF2 for degradation as shown in the legend of Fig.3.

Line 625-26:” This is illustrated by a very recent work from Song et al. wherein the association between mitochondrial metabolism and redox …”. Need to cite the reference.

Author Response

Reviewer 1:

General statement: This is a clearly-written review summarizing the roles of NRF2 and related signaling pathways that regulate and interact with mitochondrial function with a focus on malignancy and cancer stem cells. I recommend this manuscript being accepted with minor revisions as follows.

R: The authors make the points that: 1) malignant cells are pro-oxidation with high ROS and ROS-scavenging capacity (e.g., line 480-481); 2) cancer stem cells have low ROS with high tumorigenicity (e.g. Figure 6). It is also known the high ROS drive malignancy (e.g., Geou-Yarh Liou and Peter Storz, Reactive oxygen species in cancer, Free Radic Res. 2010 May; 44(5). PMCID: PMC3880197). It may be difficult for readers to reconcile these two apparently opposite features existing in cancer cells. Could the authors add some discussion about this apparent contradiction and possible ways of reconciliation?

A: We thank the reviewer for having pointed out a very interesting aspect of ROS and cancer which becomes also an opportunity to discuss about the complexity of the disease and, more in general, of the ROS biology. We have tried to elaborate some considerations, in the conclusive part of the manuscript, that in our view can help the reader to better understand the apparent contradiction of ROS and malignancy in CSCs vs non-CSCs.

R: This article talks about NRF1 as nuclear respiratory factor 1 and another NRF2 as nuclear respiratory factor 2, which is not the NRF2 covered by the review. The paper does not distinguish them with distinct notations, which can be confusing. For example, Line 239-242: “NRF1 and NRF2 (the latter also known as GA-binding protein- GABP to avoid confusion with the nuclear factor, erythroid 2-like 2), and mitochondrial transcription factor B1 (TFB1M). The NRF2-target gene heme oxygenase-1 (Ho-1 also known as Hmox1) was found to stimulate mitochondrial biogenesis via NRF2 and Akt in mouse heart”. It is not clear which NRF2 the 2nd NRF2 covered in this paragraph is.

R: This reference might be helpful. “Punctum on two different transcription factors regulated by PGC-1α: Nuclear factor erythroid-derived 2-like 2 and nuclear respiratory factor 2”

A: Thank you for this comment. We agreed with the reviewer, and we changed the text. We now refer to the nuclear respiratory factor 2 as GABP. We also added a new reference (PMID: 23597778) as suggested by the reviewer.

“(…) GABP (GA-binding protein) and mitochondrial transcription factor B1 (TFB1M). On a side note, GABP is also known as nuclear respiratory factor 2, being a commonly confused with the NRF2 (nuclear factor-erythroid-derived 2-like 2) covered in the present review (PMID: 23597778).

R: Line 93: “Akt signaling can promote the phosphorylation of NRF2 by the glycogen synthase kinase 3 beta (GSK-3β) …” This statement is confusing. It appears to suggest increased Akt signaling activity promotes NRF2 phosphorylation and thus its degradation, which is contradictory with that, without PI3K/Akt activation, GSK-3 will phosphorylates NRF2 for degradation as shown in the legend of Fig.3.

A: We thank the reviewer for having noticed this mistake which had escaped our internal revision process. We have now partially rephrased the period and made the sentence consistent with the Figure 3 of the MS.

R: Line 625-26:” This is illustrated by a very recent work from Song et al. wherein the association between mitochondrial metabolism and redox …”. Need to cite the reference.

A: Thanks for having noticed it. We have now added the reference in the main text.

Reviewer 2 Report

This review summarizes progression of NRF2 in regulating mitochondrial function and address the roles played by these in cancer cells and cancer stem cells. It would be appreciated if authors could clarify following statements and improve writing in the revised version for readers: 

1)     A couple sentences of the abstract need to be reword precisely, such as “For this reason, NRF2 has emerged as a promising therapeutic target in cancer addicted cells, paving the way for the design of natural as well as chemical inhibitors.” “Excitingly, part of the cytoprotective effects of NRF2 derives from its functional crosstalk with the mitochondrial network, which extends far beyond the mere antioxidant activity, encompassing fundamental aspects of mitochondrial homeostasis including biogenesis, oxidative phosphorylation, metabolic reprogramming, and mitophagy”

2)     Please clarify lines 472-473.

3)     Briefly describe these (Lines 182-194; lines 458-466) or remove, avoiding less relevant or basic knowledge.

4)     It is suggested to modify the Fig.5 and avoid use “CSC B”, as it is not well characterized by literature evidence.

5)     Based on summarized progresses of NRF2 and mitochondria in cancer, the perspectives should be addressed.       

Author Response

General comment: This review summarizes progression of NRF2 in regulating mitochondrial function and address the roles played by these in cancer cells and cancer stem cells. It would be appreciated if authors could clarify following statements and improve writing in the revised version for readers:

A: We thank the reviewer for his/her comments and for having appreciated our work. Here below, the point-by point response to the comments.

R:    A couple sentences of the abstract need to be reword precisely, such as “For this reason, NRF2 has emerged as a promising therapeutic target in cancer addicted cells, paving the way for the design of natural as well as chemical inhibitors.” “Excitingly, part of the cytoprotective effects of NRF2 derives from its functional crosstalk with the mitochondrial network, which extends far beyond the mere antioxidant activity, encompassing fundamental aspects of mitochondrial homeostasis including biogenesis, oxidative phosphorylation, metabolic reprogramming, and mitophagy”

A: We thank the reviewer for his/her suggestion. We have rephrased both the sentences contained in the abstract, without altering their overall meaning.

R: Please clarify lines 472-473.

A: We have split the sentence in two distinct periods, which have been partially rephrased to better clarify the content. We think that in the present form the reader will find more easy to understand the message that we wanted to deliver.

R: Briefly describe these (Lines 182-194; lines 458-466) or remove, avoiding less relevant or basic knowledge.

A: We thank the reviewer for his/her suggestion. We think that some basic information would be useful to the reader for a better understanding of the following sections within the manuscript. On the other hand, we preferred to avoid a detailed description of some processes or general notions which we felt were a bit out of the scope of the manuscript. Regarding the mentioned parts of the text, we have partially rephrased the periods to make the text smoother and easier to understand to the reader.

A: As recommended by the reviewer, we also shortened the information in the manuscript (lines 182-194, page 5) by briefly describing the main pathways involved in cellular energy metabolism, serving as a small introduction to the topic of bioenergetics.

“All cellular processes need a constant supply of energy, which can be generated by several metabolic pathways that break down complex macromolecules into intermediate metabolites to generate ATP. Two main pathways are involved in cellular energy metabolism: glycolysis and OXPHOS. Glucose is of extreme importance for cellular metabolism as it is the main source of energy for most tissues, needed to fuel both aerobic and anaerobic respiration. Through glycolysis, glucose gives origin to 2 pyruvate molecules. Those can be converted into acetyl CoA, used in the tricarboxylic acid cycle (TCA) to generate ATP along with the electron transport chain (ETC)” – lines 181, 187

R:  It is suggested to modify the Fig.5 and avoid use “CSC B”, as it is not well characterized by literature evidence.

A: We thank the reviewer for his/her suggestion. We have now removed the definition of “CSC A” and “CSC B” which actually referred to the Figure 6.

R:  Based on summarized progresses of NRF2 and mitochondria in cancer, the perspectives should be addressed. 

A: We thank the reviewer for his/her suggestion. We have now added some general conclusions and perspectives in a separate paragraph at the end of the MS entitled “Conclusions and future perspectives”